# Recycling of Post-Use Bioprocessing Plastic Containers—Mechanical Recycling Technical Feasibility

**Duc-Nam Luu** [1,2,*], **Magali Barbaroux** [3], **Gaelle Dorez** [4], **Katell Mignot** [3], **Estelle Doger** [5], **Achille Laurent** [2], **Jean-Michel Brossard** [4] and **Claus-Jürgen Maier** [6]

1 Arts et Métiers Institute of Technology, LCPI, HESAM Université, 151 boulevard de l'Hôpital, 75013 Paris, France
2 Sanofi, 82, Avenue Raspail, 94250 Gentilly, France
3 Sartorius, ZI des Paluds, Avenue de Jouques, 13400 Aubagne, France
4 Veolia Recherche & Innovation, Zone Portuaire, 291 Avenue Dreyfous Ducas, 78520 Limay, France
5 Sanofi Pasteur Lyon Carteret, 14 Espace Henry Vallée, 69007 Lyon, France
6 Sanofi, Industriepark Höchst, 65926 Frankfurt am Main, Germany
* Correspondence: duc-nam.luu@ensam.eu

**Abstract:** Most of the plastic-based solutions used in bio-manufacturing are today incinerated after use, even the not "bio-contaminated". Bioprocessing bags used for media and buffer preparation and storage represent the largest amount today. The aim of this work was to technically assess the feasibility of the mechanical recycling of bioprocessing bags. Materials from different sorting and recycling strategies have been characterized, for their suitability of further use. Quantitative physical and mechanical tests and analysis (FTIR, DSC, TGA, density, MFI, color, tensile, flexural, and Charpy choc) were performed. The data show that these recycled plastics could be oriented towards second use requiring physical properties similar to equivalent virgin materials. A comparative life cycle assessment, based on a theoretical framework, shows that mechanical recycling for end of life presents the advantage of keeping material in the loop, without showing a significant statistical difference compared to incineration with regards to the climate change indicator.

**Keywords:** mechanical recycling; single-use assembly; bioprocess industry; plastic circularity; life cycle assessment; bioprocessing bags end-of-life

## 1. Introduction

Despite its intrinsic role to provide and make available [1–3] treatments for patients, the pharmaceutical industry does not avoid the usual paradigm of all manufacturers in terms of environmental issues. Studies showed the links between GHG and the development of diseases. [4–6]. Eckelman et al. suggests that the pharmaceutical industry contributes between 4.4 to 4.6% of the worldwide Greenhouse Gas (GHG) emissions [7]. Environmental concerns of medicine include other aspects such as the ecotoxicity [8]. This could be identified in manufacturing facilities [9,10] or hospital [11,12] effluents and even in rivers [13,14]. To manage this paradox between making treatment available and affordable versus the environmental impacts that contribute to health concerns of populations, the pharmaceutical industry should embrace the holistic approach of eco-design [15].

One way to manufacture Active Pharmaceutical Ingredients (API) is through biotechnology [16]. The birth of modern bioprocess was identified in the mid-1970s with recombinant DNA [17]. Historically, the biologics were traditionally made through stainless steel (SS) infrastructure [18]. As markets were growing, a second period began in the 1990s with both an increased number of biotechnology molecules-based on clinical research and bigger facilities for industrialization [19]. With the aim to reduce cost, increase quality, speed, and gain more flexibility of production [20], Single-Use Technologies (SUTs) first appeared. Galliher et al. described a significant change around 2010 where SUTs switched

from "support systems" to "production systems" [18]. Recently, SUTs were made part of the response to the COVID-19 crisis, by allowing shorter timelines to implement production processes for vaccines [21]. Despite these advantages, the perception of SUTs remains subject to concerns [22].

The positive environmental impact of SUTs could be improved by avoiding incineration and landfills, and implementing circularity options, when possible. Because SUTs are usually composed of multilayers and multiple polymer types and because they are often treated as bio contaminated wastes, they are seen as difficult to recycle. However, according to the World Health Organization (WHO), of the total amount of waste generated by health-care activities, about 85% is general, non-hazardous waste [23]. If this assumption is confirmed for plastics in biomanufacturing, this would create recycling opportunities in this field, provided the waste management system in place would sort these two categories of waste. The recycling of these materials represents a way to contribute to this topic. Especially because the plastic used in the bioprocess industry complies with strict regulation, such as REACH for instance, and that the suppliers provide extractable guides to allow standard toxicology study for risk assessment.

The environmental burden of plastic is a topic known in the scientific community. Researchers have documented the marine pollution [24–26] and the impact on health [27–29] of plastics. In their paper, Li et al. mention that plastic pollution can be summarized as diverse, persistent, global, and threatening to human health. Furthermore, the authors support the fact that the pollution can be direct or indirect in aquatic, atmospheric, and terrestrial systems [30]. The proper management of plastic waste represents a key issue [31], which is taken into consideration increasingly more by international organizations, such as the European Union [32].

An optimized eco-design approach requires anticipating the end of life of the product. The circularity of products and materials (at their highest value)—"in practice" at scale and not only from a theoretical or laboratory level—is one of the key principle of the circular economy [33]. After tracking, collection, and sorting, the technical requirements for recycling must be met. As an example, for the packaging, the ISO 18604:2013 [34] is mentioning "reprocessing, by means of a manufacturing process, of a used packaging material into a product, a component incorporated into a product, or a secondary (recycled) raw material; excluding energy recovery and the use of the product as a fuel".

The EU commission sorts plastic recycling in three categories: mechanical, chemical, and organic recycling. Organic recycling is defined by the EU Packaging and Packaging Waste Directive 94/62/EC as aerobic treatment (industrial composting) or anaerobic treatment (bio gasification) of packaging waste and is out of this study scope. Mechanical recycling refers to operations that aim to recover plastics via mechanical processes (grinding, washing, separating, drying, re-granulating, and compounding). For mechanical recycling, the ability to separate and sort components and materials in post-used products must be taken into consideration to confirm that the quality of the post-used materials meets the recycling process specifications and expectations as a secondary raw material in a new use. Some materials or substance incompatibilities can affect the success of mechanical recycling with the current technologies. To overcome mechanical recycling limits, chemical recycling by dissolution/reprecipitation, thermocatalytics, steam cracking, or pyrolysis [35,36] have been widely investigated. However, the scalability at industrial level and its environmental benefit thereof still needs to be validated, especially from a lifecycle perspective [37]. Many technical parameters such as the massification of the feedstock quantity, the quality maximized, the olefin recovery yields (compared to ethylene yields from naphtha for example), and purification of the pyrolysis oil must be challenged to validate efficiency and robustness of such process. The existing pilots still require the sorting of plastics [37,38]. At the moment, lack of data for life cycle assessment makes the overall environmental impact evaluation of the process difficult and the dissolving rates of the recycled stream in the refinery is a traceability challenge for allocating the recycled content. Chemical recycling is an additional set of technologies to recycle plastics which

cannot be mechanically recycled [39] and mechanical recycling in standard waste streams, when possible, is on top of the recycling hierarchy.

The driver for this study was to challenge the widespread thought that SUTs in general cannot be recycled in standard waste streams for plastics. Therefore, we asked ourselves if the mechanical recyclability of the Single-Use Technology faces a technical constraint and focused the paper to address some answers.

The inclusion of stakeholders all along the value chain of a product, such as the supplier, end-user, and recycler, is a key factor of an eco-design approach [36]. With this aspect in mind, a collaboration between Sartorius, Sanofi, and Veolia was established to explore if, when carefully designed, plastic parts used in bioprocessing for non-hazardous application can technically be diverted from incineration or landfill and recycled into raw materials with the quality equivalent to pristine grades of polymers.

First, the sorting and manual separation could be managed at the industrial sites producing the waste and recirculation routes already existing in practice for the main component materials (mainly LDPE). Additionally, a mechanical extrusion process exists at scale. Nevertheless, the material separation strategy and the quality of the secondary raw material that could be recovered from this end-of-life product have to be evaluated. Results obtained at laboratory scale on representative items will then serve the discussion on the different recovery and valorization scenarios including criteria of logistics, volume, and uses.

The focus of this work was to identify and analyze the product separation strategy, to characterize the quality of the recycled plastic at a laboratory scale on representative items. To engage the discussion and to evaluate if a circular option can be an environmentally sustainable solution, we estimated the environmental footprint, based on data available in the literature. These results will then serve a future study on the different recovery and valorization scenarios including criteria of logistics, volume, and uses.

## 2. Single-Use Technology

Before the SUT recycling possibilities are discussed, we define what we refer to when talking about SUT material in this paper.

Single-Use Technologies (SUTs) refers to technical solutions based on Single-Use Assembly (SUA) which is also called Single-Use Systems (SUS). An SUA is an engineered process equipment solution most commonly assembled from components made of polymeric materials, which together create a system or unit operation designed for one time of campaign use [40]. In other words, an SUA is a ready to use, closed processing equipment consisting of integration and presterilized components. These systems are usually sterilized by gamma irradiation, but X-ray is today emerging. A sterile bioreactor composed of plastic parts such as bag, impeller, spargers, tubes, and connectors, is an example of an SUA. A storage bag with or without filter, or a mixing bag with or without single-use sensors with all parts being made of plastic materials, are other typical examples.

These systems are said to be single use because, when compared to conventional stainless-steel equipment, they are not designed nor qualified to be cleaned and reused for another campaign. However, the duration of use might reach weeks, months, and even years for long-term storage application in cold or frozen states (e.g., bulk storage before final form and fill).

Surveys suggest that waste to energy is currently the most widely used post-use management method within the biopharmaceutical industry [41]. Indeed, incineration is often the only possible option for the treatment of what is considered as a bio-contaminated waste.

Rogge et al. proposed a comparison between SS plants and SUA ones. They suggest that for most of the parameters, SUA presents more advantages in terms of economical and quality aspects [42]. Whitford et al. performed a life cycle assessment (LCA), showing less environmental burden of SUA versus SS as shown in Figure 1. This can be explained by the reduced energy and water consumption due to the elimination of cleaning and sanitization.

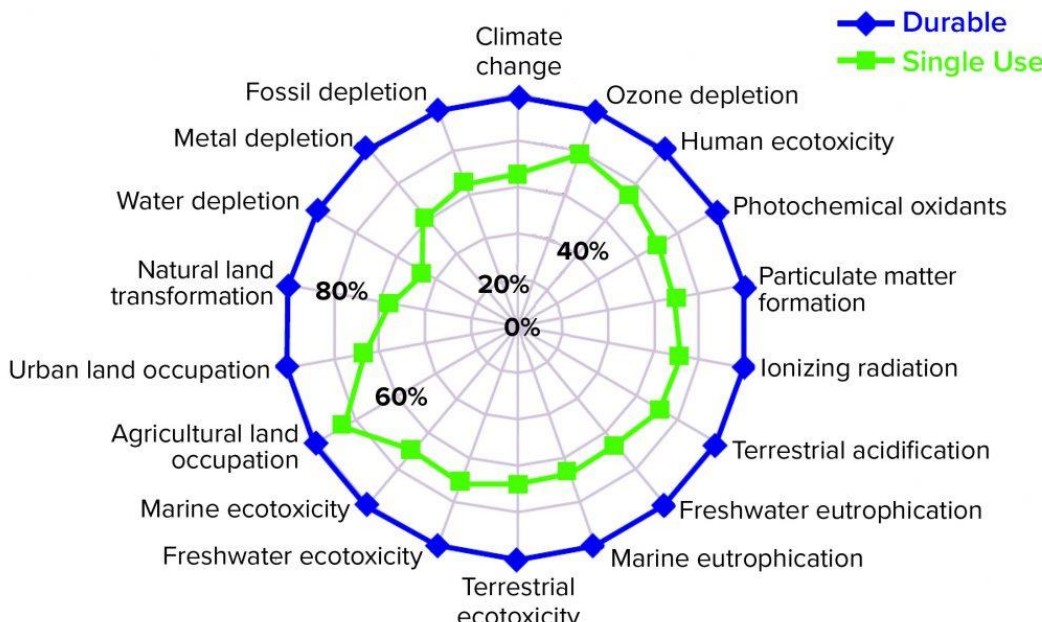

**Figure 1.** On average, SU facilities are more ecofriendly than traditional (durable) facilities in 18 distinct categories of environmental pressure by Barbaroux et al. [43].

In addition to these elements, SUT presents the advantage of avoiding investments related to the installation of the infrastructure which supports SS [44]. This leads to both time and economical savings. By easing technical and regulatory scale up, SUT can significantly simplify the processes of industrialization by just integrating the production unit into modular clean rooms [45]. This last point represents a major advantage when it comes to the need to make new treatment for patients available as soon as possible. The success of making the COVID-19 vaccine available in one year is a relevant illustration of this aspect [21].

The aim of this collaboration was to revisit the current practices of post-use management of SUA to assess alternatives to traditional end-of-life methods. The first step was to identify which SUAs need to be treated as hazardous waste and which do not. For example, SUA has been initially used for buffer and media preparation and storage in aseptic mixers and bags [46]. Due to the composition of standard buffer and media, these SUAs may not be required to be handled as hazardous waste. This segregation between hazardous and not hazardous opens the door to alternative post-use management options such as sorting and recycling, which is the focus of this article.

## 3. Materials and Methods

### 3.1. Materials

In this study, three different finished products in standard double packaging have been analyzed. Materials for the constitution of SUA are well known and characterized. They are compliant with regulatory requirements, such as REACH (EU Registration, Evaluation, Authorization and restriction of Chemicals) for instance. Quantities and chemical characteristics of the additives added to polymers were optimized to be compliant with bioprocess applications [47].

The transparent film which constitutes the bag chamber (Figure 2) for storage bags, mixers, or bioreactors is mainly composed of low-density polyethylene (LDPE), linear low-density polyethylene (LLDPE), and polyethylene vinyl alcohol (EVOH) as a barrier layer [48]. EVOH is widely used in multilayer structures for its excellent flex-crack resistance, high resistance to hydrocarbons, oil, solvent, and for its gas barrier properties. Since it is sensitive to moisture, it is recommended to use it between at least two layers of a

hydrophobic material, such as LDPE [49,50]. In this film the ratio of EVOH is such as it has a minor impact on the recycled material [51].

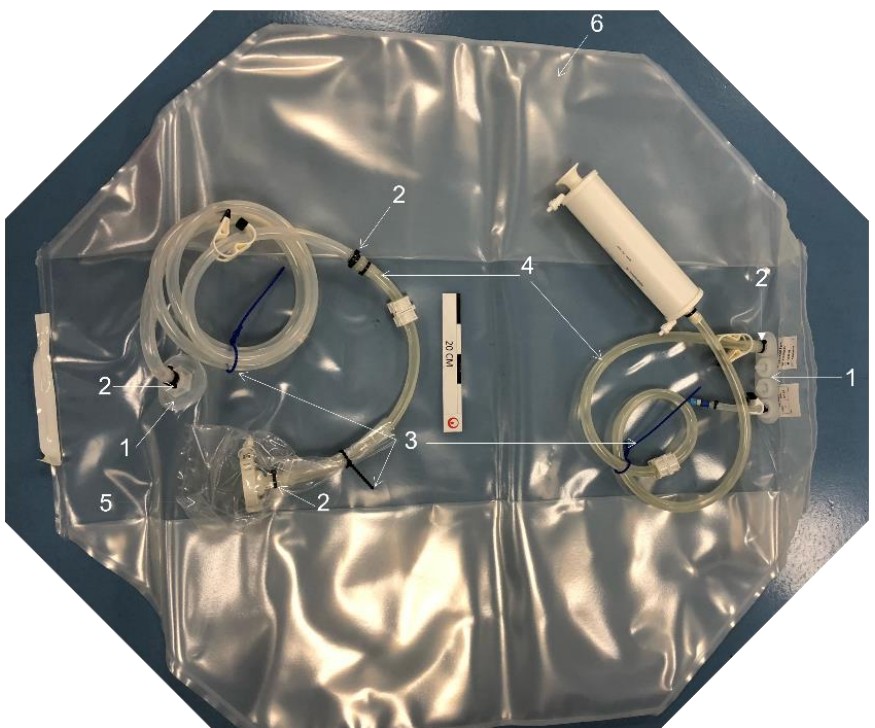

**Figure 2.** Example of the structure of a bag: 1—cable tie, 2—bag port (welded on the film), 3—packaging tie, 4—tubes, 5—packaging overwrap, 6—film-bag chamber is composed of the film and the welded flange ports.

Packaging film is transparent and composed of LDPE as a sealant layer to guarantee barrier to sterility and polyamide (PA) for its good gas barrier properties [52], its exceptional mechanical strength, high resistance to impact, and puncture and pin holding [53] which are expected properties for packaging of sterilized device. PA in multilayer films is currently classified as non-recyclable in many design guidelines but this indeed depends on PA type and concentration.

Tubes are translucent, composed of a synthetic thermoplastic elastomer suitable for pumping application, and connected to the bag with black polyamide cable ties.

In order to evaluate impact of dismantling/sorting strategy on quality of recycled product, two grades of rLDPE have been prepared in this study from post-industrial SUA:

- The first one, "rLDPE_Pure", represents a total dismantling with complete and clean disconnection of the tubes at the port flange. Thus, this first material is composed of the bag chamber, itself composed by the film and port flanges (Figure 2).
- The second one, "rLDPE_Blend", represents a rapid dismantling without disconnection of the tubes and cable ties, but with a cut (or a sealing equipment for a totally closed disconnection) at the port flange level. Thus, this second material is composed of bag chamber (film and port flanges), cable ties, pieces of tubes, and additionally, packaging (Figure 3). All the non-PE components can thus be considered as "foreign" materials in the "rLDPE_Pure" stream.

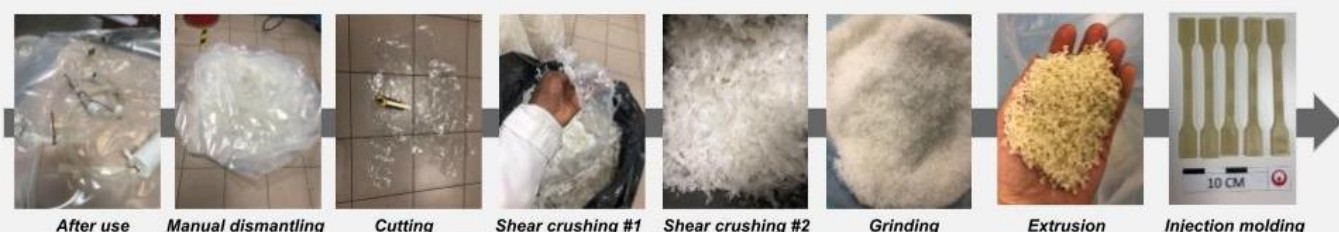

**Figure 3.** Steps of material preparation of rLDPE_Pure.

- A cable tie (1) is a "fastener" as described in the *ISPE Good-Practice Guide: Single-Use Technology* [54], page 17: non-wetted components used to keep the tubing (4) in position of the fitting or port (2), with adjustability to variable circumstances. These cable ties are fixed with calibrated equipment, they are not intended to be removed after installation.
- Some ties (5) are used to maintain packaging, manually assembled to wrap, and protect sensitive components during shipping, to be removed manually before use.

Plastic cable ties are today widely used in bioprocessing application, mainly in polyamide and black color, but there is an emerging market requesting different colors for lines identification and differentiation.

Cable tie is a critical component, as it is a key element for engagement safety and as such is part of the extensive qualification process for the integrity of single assemblies and must be therefore carefully selected.

Metal clamps could potentially represent an option that would allow for an easier sorting at end of life.

### 3.2. Samples Production

#### 3.2.1. Material Preparation

The SUA followed several steps of preparation to achieve the two grades of materials. Firstly, all parts were manually disconnected. Every element was grinded individually.

For the "rLDPE_Pure", the mainstreams (bag chambers) were shredded two times with a shear crusher and finally grinded with a knife grinder, with a mesh of 5 mm size (Figure 3).

For the "rLDPE_Blend", the secondaries streams (ties, pieces of tubes, packaging) were individually grinded with a knife grinder with a 4 mm mesh size.

#### 3.2.2. Extrusion

The "rLDPE_Pure" material was directly extruded, from the flakes stream composed by bag chamber and port flange, in a co-rotating twin-screw extruder and pelletized with a rotary cutter (Figure 3). The extrusion temperature profile was in the range 190–235 °C and the rotation speed of the screw was 350 tr/min. A filter $80 \times 40$ mesh was used at the end of the die to avoid unmelted impurities into the recycled granulates.

The "rLDPE_Blend" was obtained by blending the flakes of all the different components. To be representative of the proportions of each material in the different SUAs, a quantitative analysis of the mass of the different parts of the bag chamber was carried out (Table 1). The choice of blending formulation was based on a worst-case scenario, i.e., the SUA with the highest ratio of "foreign" material such as the lower (bag chamber/total) weight ratio. The SUA1 was the one with the highest ratio. After that, all the flakes from the different components were mixed and homogenized with the following proportion: 55% bag chamber/port flange, 42% packaging, 0.5% ties, and 2.5% pieces of tubes. The resulting flakes were extruded with the same condition as the "rLDPE_Pure" material.

**Table 1.** Composition of different SUA.

| Streams | SUA1 | SUA2 | SUA3 |
|---|---|---|---|
| Bag chamber | 372 g (55%) | 565 g (66%) | 1546 g (59%) |
| Packaging (plastic film) | 290 g (42%) | 261 g (31%) | 1044 g (40%) |
| Ties | 3 g (0.5%) | 4 g (0.5%) | 5 g (0.2%) |
| Pieces of tubes * | 16 g (2.5%) | 20 g (2.5%) | 15 g (0.7%) |
| Total | 681 g | 850 g | 2610 g |
| Weight ratio (bag chamber/total) | 54.6% | 66.4% | 59.2% |

* The mass of the pieces of tubes correspond to the weight of the tube left on the port flange after that the tube has been cut without removing the ties.

### 3.2.3. Injection

For the mechanical and colorimetric properties measurements, standard samples were molded by injection using an injection molding device with a clamping force of 450 kN. This device is equipped with a screw of 25 mm diameter and an effective screw length (L/D ratio) of 24. For the two batches, the temperature range was 200–240 °C and the mold temperature was set at 40 °C. The injection mold is composed of one tensile bar type 1A according to ISO 527-2:2012 [21], one bar $80 \times 10 \times 4$ mm$^3$ for flexural and Charpy impact test and one disk with a thickness of 1 mm and a diameter of 50 mm for colorimetric analysis.

### 3.3. Methodology and Testing

### 3.3.1. Fourier Transform Infra-Red (FTIR)

Fourier transform infra-red were recorded between 4000 and 650 cm$^{-1}$ using a Spectrum 100 set us in ATR mode (4 scans, resolution of 4 cm$^{-1}$) on the four streams.

### 3.3.2. Differential Scanning Calorimetry (DSC)

The melting and crystallization behaviors of the four streams of the SUA were determined using a DSC under nitrogen atmosphere. The analysis conditions of melting temperature measurement depend on the piece of the SUA considered. For the bag chamber and port flange, the samples were cooled down to −30 °C, held at isothermal for 8 min, heated up to 200 °C (first cycle of fusion), then cooled down again to −30 °C at 10 °C/min, held at isothermal for 8 min (cycle of crystallization), and heated up to 200 °C again at 10 °C/min (second cycle of fusion). The packaging followed the same protocol with a range of temperature from −30 to 300 °C. For the ties, the sample was heated up to 350 °C at 10 °C/min, held at isothermal for 2 min, then cooled down to −25 °C at 10 °C/min, and finally heated up again to 300 °C with one scanning temperature rate of 10 °C/min. The thermograms presented in this study focus on the cycle of crystallization and the second cycle of fusion, to disregard the thermal history of the material.

### 3.3.3. Macro TGA

A thermogravimetric was used to analyze the thermal degradation. Sample weights at around 4.5 g were heated up from 40 °C to 750 °C at a heating rate of 10 °C/min under laboratory atmosphere followed by 3 h temperature dwell at 750 °C. Only the residual weight at 750 °C was determined from this analysis. Three measurements were applied for both "rLDPE_Pure" and "rLDPE_Blend" materials.

### 3.3.4. Density

A densitometric balance was used to determine the volumetric mass in water at 23 °C, according to the standard ISO 1183-1:2012 [55].

### 3.3.5. Melt Flow Rate

The melt flow rate is an indirect measurement of the melt viscosity of materials. The MFR measurements were carried out in a melt flow index, at 190 °C, 2.16 kg, according to the standard NF EN ISO 1133-1:2011 [56].

### 3.3.6. Color

A spectrocolorimeter was used to determine the three-color components in the CIE $L^*$, $a^*$, $b^*$ domain, with the parameters: SCE, Illuminant D65, angle 10°. A method to determine the opacity was implemented by measuring the $L^*$, $a^*$, $b^*$ of a 1 mm thickness sample on a white and a black background. The $\Delta E$ will be calculated with the formula:

$$\Delta E = \sqrt{\left(L^*_{white} - L^*_{black}\right)^2 + \left(a^*_{white} - a^*_{black}\right)^2 + \left(b^*_{white} - b^*_{black}\right)^2}$$

$L^*$ correspond to the lightness axis ($L^*$ = 100 for the white and $L^*$ = 0 for the black), $a^*$ correspond to the green-red axis and $b^*$ correspond to the blue-yellow axis.

### 3.3.7. Tensile Tests

The tensile tests were carried out according to NF ISO 527-1:2012 [57] using a type 1A tensile bar on a tensile test machine with a 10 kN load cell. The crosshead was moved with a constant velocity of 50 mm/min for strength and the strain at break determination. The Young modulus was determined with a constant velocity of 1 mm/min and an extensometer. At least five measurements were obtained for each test and material. The results are presented in Supplementary File S1.

### 3.3.8. Flexural Test

The flexural tests were carried out according to NF EN ISO 178:2010 [58] using bars 80 × 10 × 4 mm$^3$ placed on a three-point flexural device with a 5 kN load cell adapted on a tensile test machine. The crosshead was moved with a constant velocity of 2 mm/min and the flexural modulus was determined on the average of five measurements.

### 3.3.9. Charpy Choc

The Charpy impact tests were carried out according to NF EN ISO 179-2:2010 [52] using a test specimen type 1 (80 × 10 × 4 mm$^3$), a type of notched A and a 5J hammer on a Charpy impact test device.

## 4. Results

In general, most of the plastics contain additives (flame retardant, pigments ... ) which are needed to meet technical requirements of a product during its first life. However, these additives could create hurdles during the recycling process and limit second life options for recycled material. Therefore, a chemical analysis is usually required to define the composition of the recycling streams and anticipate potential problems of miscibility or the un-melted phase during the transformation phase. In this study, chemical analysis was used to confirm the known composition of the studied SUAs and identify potential unexpected issues.

The analyses of the FTIR spectra allow to determine the nature of the chemical bonds of the material of each component. The FTIR spectrum of the bag chamber (Figure 4) shows a characteristic spectrum of polyethylene with two peaks at 2915 cm$^{-1}$ and 2849 cm$^{-1}$ corresponding to the C-H strength bond. The presence of the wavelength at 1377 cm$^{-1}$ corresponds to CH3 bend peak and allows to assert that the polyethylene is a LDPE or a LLDPE [53]. The two faces of the bag chamber were analyzed and have the same spectra. If the bag chamber is a multilayer system, the inner layer or layers cannot be detected with the FTIR. The DSC analysis allows the characterization of all the crystalline polymers present in the layers. The bag chamber thermogram, Figure 5a, shows three melting temperatures at

109 °C, 126 °C and 186 °C perfectly aligned with the literature for LPDE (Tmelt1 = 109 °C), LLDPE (Tmelt2 = 126 °C) and EVOH (Tmelt3 = 186 °C) (Table 2) [59–61].

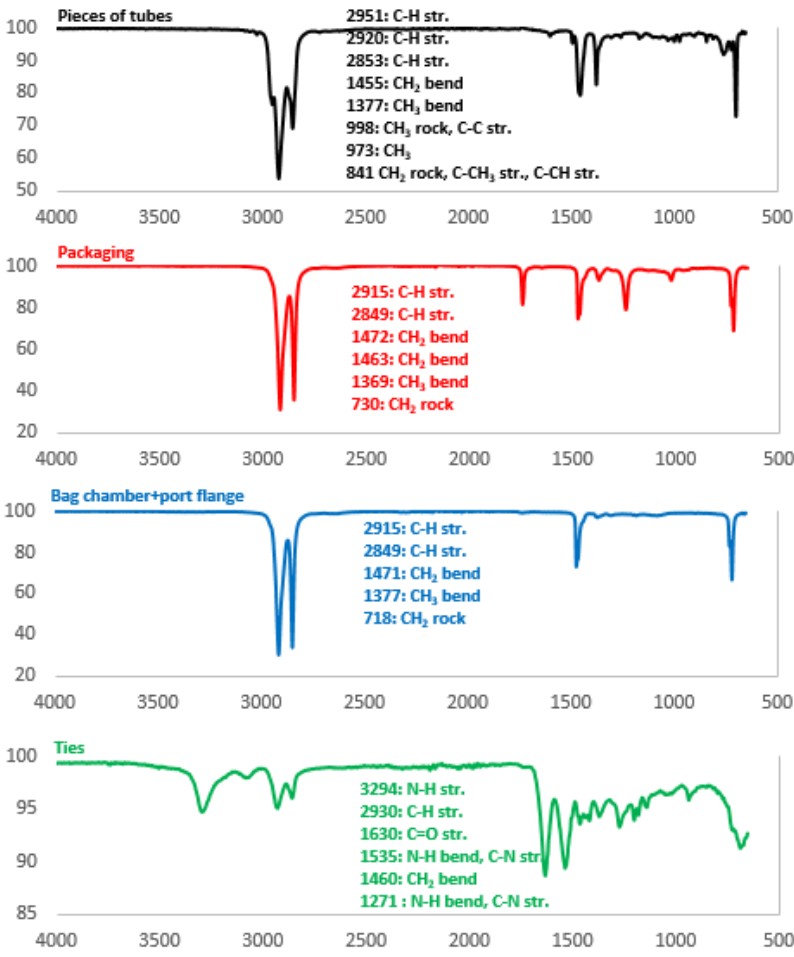

**Figure 4.** FTIR spectra of the pieces of tubes (black), packaging (red), bag chamber (blue); and ties (green).

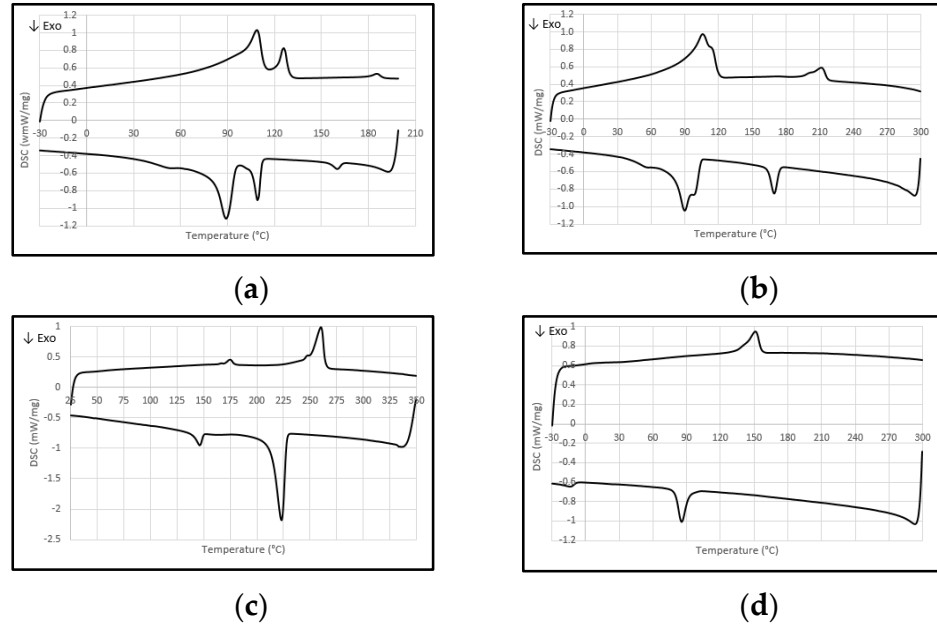

**Figure 5.** DSC thermograms of (**a**) bag chamber; (**b**) packaging; (**c**) ties; (**d**) pieces of tubes.

**Table 2.** DSC Results.

|  | Bag Chamber + Port Flange | Packaging | Tie | Tubes |
|---|---|---|---|---|
| Melting temperature (°C) | $T_{melt1}$ = 109 °C $T_{melt2}$ = 126 °C $T_{melt3}$ = 186 °C | $T_{melt1}$ = 105 °C $T_{melt2}$ = 211 °C | $T_{melt1}$ = 175 °C $T_{melt2}$ = 259 °C | $T_{melt1}$ = 152 °C |

The packaging FTIR spectrum (Figure 5b) also presents the two characteristic bonds at 2915 cm$^{-1}$ and 2849 cm$^{-1}$ corresponding to a polyethylene but additional bands relative to other polymers are detected. The packaging thermogram (Figure 6b) presents two melting temperatures: 105 °C and 211 °C (Table 2) corresponding respectively to LDPE and a PA polymer [58].

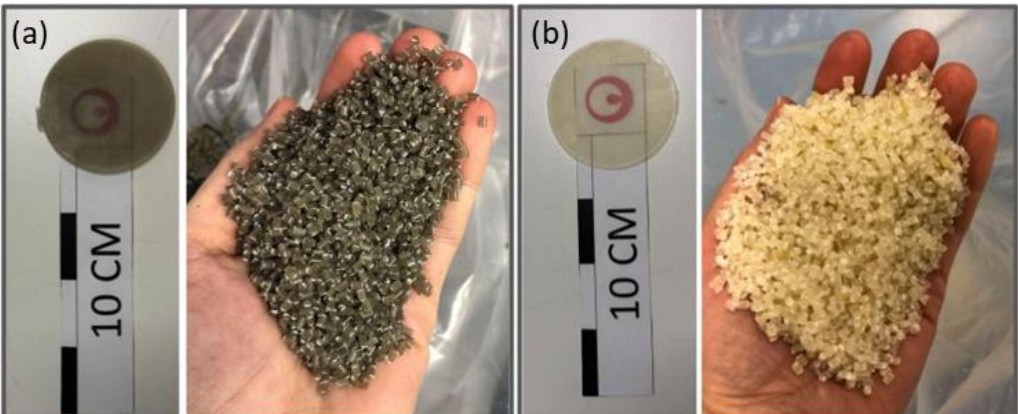

**Figure 6.** Pellets and disk of (**a**) rLDPE_Pure grade, (**b**) rLDPE_Blend grade.

Figure 5c shows the spectrum corresponding to the ties. As expected, it is close to a polyamide spectrum with several N-H and C-N bonds (3294 cm$^{-1}$, 1535 cm$^{-1}$, 1271 cm$^{-1}$) [52]. The DSC thermogram ties also present two melting temperatures at 175 and 259 °C (Table 2). These two temperatures are characteristic of the melting temperature of polyamides [61], which is consistent with the FTIR analysis.

The piece of tube FTIR spectrum (Figure 5d) presents several peaks which indicate that the plastic is composed of ethylene groups, but it is not a polyethylene or a polypropylene. To complete, the DSC analysis highlights one melting temperature at 152 °C (Table 2).

It is expected that PA and TPE may have an impact on the extrusion step of "rLDPE_Blend" grade properties and should be closely monitored.

After extrusion and injection of the two recycled grades, typical physical properties such as density, ash content, and melt flow index were measured as shown in Table 3. No significant difference between the two grades was observed. For both grades, the density is in the range of density of a virgin LDPE between 910 and 940 kg/m$^3$. The ash content is very low which means that the streams are lightly filled and the MFR are very low too.

**Table 3.** Density, ash content, and melt flow index results.

|  | rLDPE_Pure | rLDPE_Blend |
|---|---|---|
| Density (kg/m$^3$) | 927 ± 2 | 945 ± 5 |
| Ash content | 0.37 ± 0.005 | 0.38 ± 0.05 |
| MFR * (g/10 min) | 0.25 ± 0.05 | 0.16 ± 0.05 |

* MFR = melt flow rate.

The main differences between the two samples are the colorimetry and the opacity. The first observation showed that the "rLDPE_Pure" sample yellowed although the initial

flakes of the bag chamber were totally translucid, without color. After the extrusion step, the pellets present a yellowish tinge with a b* = 7.76 (corresponding to the blue-yellow axis) (Table 4) and a decrease in the opacity (Figure 6a). This decrease in properties could be due to the presence of the EVOH, which was thermo-oxidized during the extrusion phase at 190–235 °C [62]. The second observation showed the "rLDPE_Blend" grade was darker than "rLDPE_Pure" (decrease of $L^*_{LDPE_{BC}} = 78.7$ *to* $L^*_{LDPE_{WC}} = 54.7$) due to the presence of the black cable ties in the blend (Figure 6, Table 4). The transparency was also impacted with a decrease in the $\Delta E$ for 31 to 15 between "rLDPE_Pure" and "rLDPE_Blend". Change of color and an issue for a second life of the materials when they contribute to the functionality was found.

**Table 4.** Spectrocolorimeter results.

|  | rLDPE_Pure | | rLDPE_Blend | |
|---|---|---|---|---|
|  | **White** | **Black** | **White** | **Black** |
| L* (ET) | 78.7 (0.52) | 48.4 (0.68) | 54.7 (0.72) | 42.2 (0.68) |
| a* (ET) | −1.66 (0.11) | −2.71 (0.09) | 0.77 (0.03) | −1.28 (0.06) |
| b* (ET) | 7.76 (0.25) | 0.70 (0.23) | 11.94 (0.19) | 4.00 (0.28) |
| ΔE | 31.13 | | 14.81 | |

It is difficult to observe the impact of the EVOH on the LDPE, because film without EVOH has not been characterized so far. Nevertheless, EVOH in the "rLDPE_Pure grade" does not lead to a premature break of the sample during tensile tests (Figure 7).

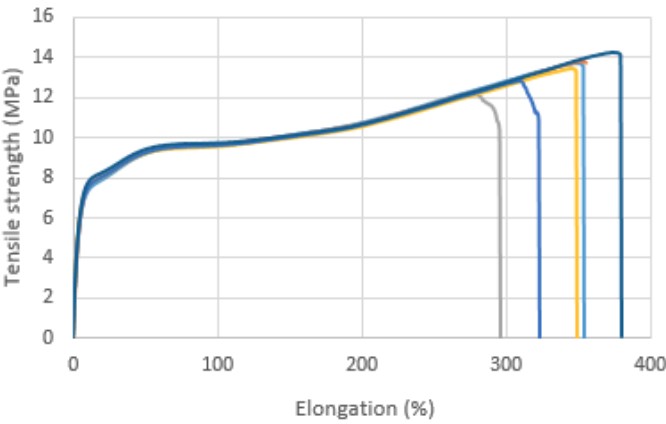

**Figure 7.** Tensile test of rLDPE_Pure grade where each color represents one sample.

When compared to "rLDPE_Pure" grade, the "rLDPE_Blend" grade shows a reduction of the elongation at break (Figure 8) and a slight increase of the young modulus while no significant difference is observed for the other parameters (Table 5). Charpy impact notched value is quite high and representative of LDPE, impact of PA addition from packaging is not visible in the "rLDPE_Blend" sample.

**Table 5.** Mechanical test results.

|  | rLDPE_Pure | rLDPE_Blend |
|---|---|---|
| Tensile stress at break (MPa) | 12.5 ± 0.7 | 11.8 ± 0.05 |
| Elongation at break (%) | 334.1 ± 13 | 217.9 ± 3 |
| Young modulus (MPa) | 200 ± 5 | 231 ± 1 |
| Flexural modulus (MPa) | 178 ± 7 | 179 ± 3 |
| Charpy impact notched (kJ/m$^2$) | 48.5 ± 0.75 | 47.6 ± 1.8 |

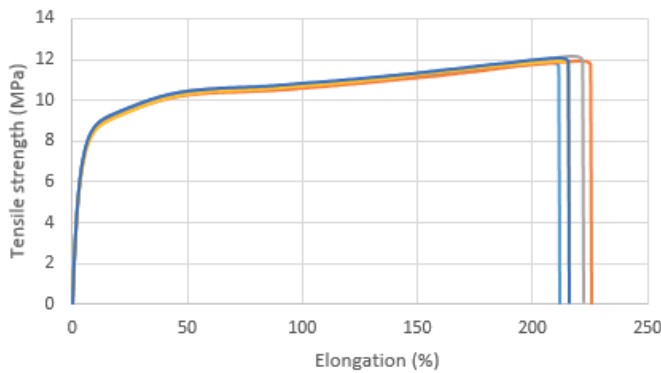

**Figure 8.** Tensile test of rLDPE_Blend grade where each color represents one sample.

## 5. Discussion

The overall purpose of this research is to reduce the environmental impact of the end-of-life of post-use bioprocessing plastic containers. Therefore, the mechanical technical recyclability of SUA will be discussed and the environmental profiles of different end of life scenarios, will be compared, using a comparative LCA approach.

### 5.1. Mechanical Technical Recyclability

Simulation of the mechanical recycling stages performed in this study showed that main mechanical recycling steps such as extrusion and injection are compliant with industrial recycling practices.

Different strategies of dismantling and blending can be considered regarding the end of life of industrial items such as SUAs. In the first approach, "rLDPE_Pure favors the purity of the LDPE stream by dismantling and carefully sorting each component material. The second approach, "rLDPE_Blend", minimizes the effort for waste logistics and maximizes the amount of one unique recycled LDPE grade containing traces of "foreign" materials.

Technical evaluation of these two options shows comparable properties, close to virgin LDPE and to existing LDPE recycling grades (Tables 6 and 7, Figure 9). Thus, it is assumed, as a first approach, that both streams can be oriented to application with similar performance requirements than the one of virgin material (films, bags, pipes and fittings, profiles, and flexible sheets).

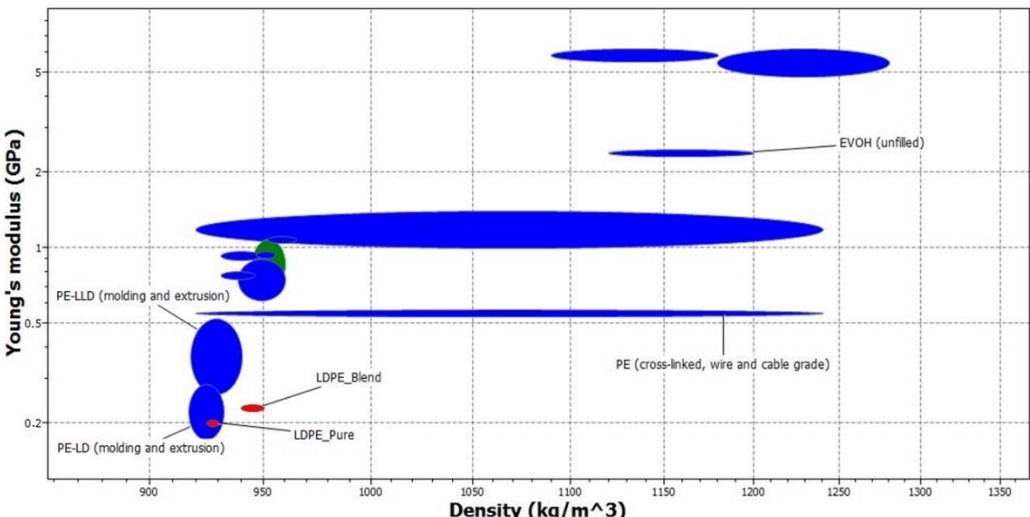

**Figure 9.** Tensile test of rLDPE_Blend grade.

**Table 6.** Corresponding rate with commercial LDPE grades (virgin LDPE_V or recycled LDPE_R) by applying a mean square calculation based on the different properties measured.

|  | LDPE_V | LDPE_R |
|---|---|---|
| rLDPE_Pure | 74% | 92% |
| rLDPE_Blend | 72% | 92% |

**Table 7.** Comparison of physical and mechanical properties of rLDPE_Pure and rLDPE_Blend grades produced versus virgin and commercial recycled grade.

| Properties | Test Method | rLDPE_Pure | rLDPE_Blend | LDPE_V [1] | LDPE_R [2] | Unit |
|---|---|---|---|---|---|---|
| Density | ISO 1183 | $927 \pm 2$ | $945 \pm 5$ | 932 | 915–970 | $kg/m^3$ |
| Melt flow rate | ISO 1133 (190 kg; 2.16 kg) | $0.25 \pm 0.05$ | $0.16 \pm 0.05$ | 0.8 | 2–5 | g/10 min |
| Young module | ISO 527 | $200 \pm 5$ | $231 \pm 1$ | 620 | 199 | MPa |
| Tensile stress at yield | | $13.4 \pm 0.37$ | $11.4 \pm 0.37$ | 18 | 10.1 | MPa |
| Tensile stress at break | | $12.5 \pm 0.7$ | $11.8 \pm 0.05$ | N.R. | N.R. | MPa |
| Elongation at break | | $334.1 \pm 13$ | $217.9 \pm 3$ | 700 | 75 | % |
| Flexural modulus | ISO 178 | $178 \pm 7$ | $179 \pm 3$ | N.R. | N.R. | MPa |
| Charpy notched | ISO 179 | $48.5 \pm 0.75$ | $47.6 \pm 1.8$ | 30 | 30 | $kj/m^2$ |
| Ash content | ISO 3451-1 | $0.37 \pm 0.005$ | $0.38 \pm 0.5$ | 0 | N.R. | % |

[1] SABIC ® LLDPE 6135BE—used for extrusion. [2] A commercial recycled grade.

However, when compared to the "rLDPE_Pure" grade, the "rLDPE_Blend" grade shows a lower elongation at break (35% lower), a higher Young modulus (15.5% higher), and grayish and less transparent color—which could limit potential applications.

Although there exists a wide range of fields of application for mechanically recycled LDPEs with similar high properties i.e., bags and industrial films, packaging and containers, and construction materials, this material will neither achieve medical grade nor even food grade compliance, as no mechanical extrusion process is currently validated by EFSA to deliver an LDPE food grade certificate. This study focused on post-industrial recycled (PIR) material, and it will be necessary to include, in the case of post-consumer recycled (PCR) material, at least one washing step before the dismantling step and possibly drying step after grinding. As it represents a usual standard step in Polyolefin mechanical recycling, no major challenge is expected.

*5.2. The Strategy for Recycling*

Due to the low use quantities, the "purest" approach requires the dismantling and sorting, organized as a manual post-use operation, on site. Indeed, the low quantities of post-used products will require mixing with other waste streams, assisted by automation, with the risk of downgrading the quality of the outcome. Therefore, the main goal will be to massify all these products to avoid the blending with other LDPE sources for which traceability could not be guaranteed, contrary to the healthcare products.

Because of the formulation, the "blended" option will likely face challenges to identify a suitable valorization route. Application opportunities of this grade will likely be limited due to the dark color, from the black cable ties, even in small quantity (0.5 wt.%). Short-term recommendations shall be to eliminate this component from the blending to maintain the light color. A longer-term option because it is extremely resource intensive, could be a re-design and qualification on an alternative connection.

### 5.3. Environmental Profiles of Different End of Life Scenarios

A comparative LCA was performed to compare the chemical recycling vs. mechanical recycling vs. incineration with energy recovery by considering 1 kg of "rLDPE_Blend" treated. The methodology is presented in Supplementary File S2. The overall scope is presented in Supplementary File S3 and the main assumptions and life cycle inventory are presented in Supplementary File S4. As incineration with energy recovery is the current waste stream, it is considered as the reference scenario. The mechanical and chemical recycling are considered as two different scenarios. As shown in Figure 10, the incineration seems to present a similar profile than the mechanical recycling of "rLDPE_Blend" for the climate change short term (GWP100 method) impact. Nevertheless, as illustrated for the mineral resources use, the mechanical recycling for the "rLDPE_Blend" seems to have a better profile for most of the indicators compared to the incineration and chemical recycling.

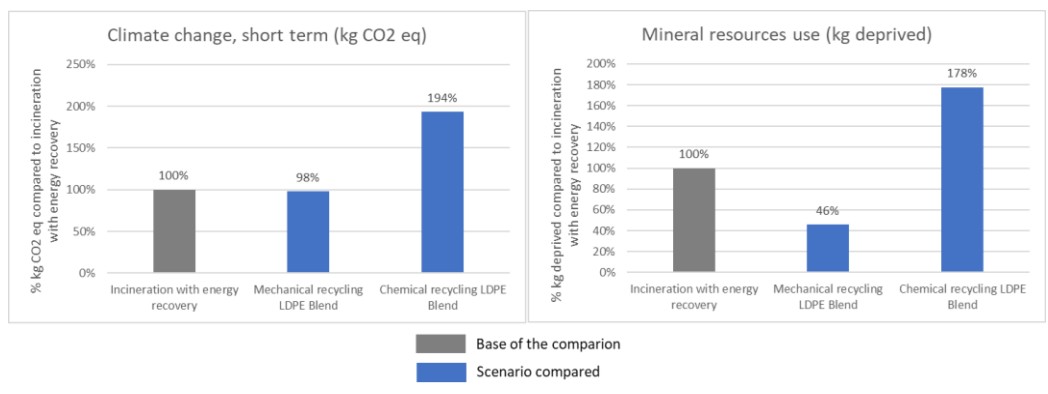

**Figure 10.** Comparative LCA results of mechanical treatment vs. incineration vs. chemical treatment for 1 kg of SUA treated, for two mid-point indicators, climate change short term and mineral resources use (IMPACT World + Midpoint V1,01).

A sensitivity analysis is proposed by assessing "rLDPE_Pure" and "rLDPE_Blend". The overall results are disclosed in Supplementary File S5. If we focus on mechanical and chemical treatments, results show that for 1 kg of recycled LDPE in the output of the processes, all mid-point indicators are in favor of mechanical recycling, as illustrated in Figure 11.

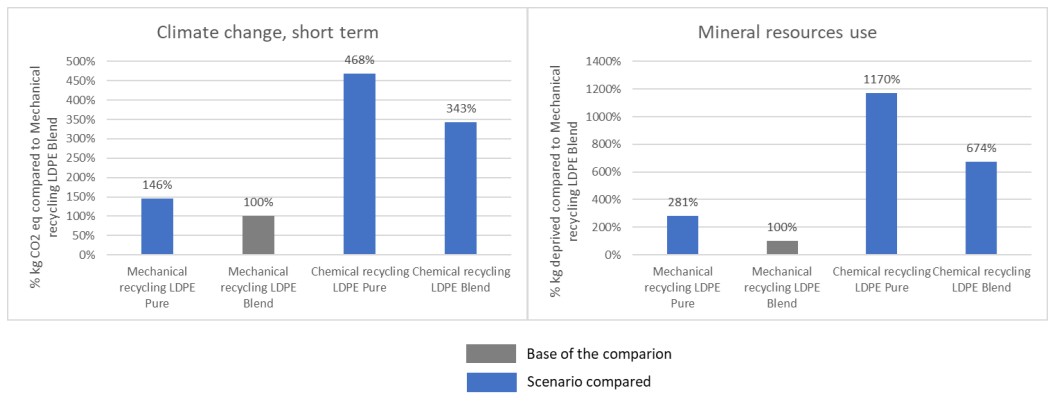

**Figure 11.** Comparative LCA results of mechanical treatment vs. chemical treatment by considering "rLDPE Blend" and "rLDPE Pure" for 1 kg of LDPE recycled, for two mid-point indicators, climate change short term and mineral resources use (IMPACT World + Midpoint V1,01).

From the different data generated by the LCA, we can assume that the chemical recycling has the worst environmental footprint. The mechanical recycling is not worse than the incineration for most of the environmental aspects. However, the calculations were performed based on literature data for all types of recycling. As chemical recycling is still in the pilot

phase, the data and hypothesis related to this process remain uncertain, unlike the incineration and the mechanical recycling which are well implemented. The chemical recycling presents the advantage to provide a virgin-like material and choice of material between chemically or mechanically recycled material should be discussed in terms of balance between final use property requirements, targeted environmental benefits, and costs.

In a circular perspective, mechanical and chemical recycling present the advantage of keeping the material in the loop in contrast to incineration which is not circular even with energy recovery [33]. Main difference between chemical and mechanical recycling is linked to the recycled material in output, and also to the readiness level of chemical recycling technologies thus to environmental data available at scale. Depending on the requirements, the choice of a material will be based on its performance. In an eco-design perspective, such material selection should be decided at early design stages based on both their technical characteristics and their environmental footprint.

## 6. Conclusions

Through this study, we highlighted that recycled plastics from Single Use Assembly (SUA) for bioprocessing could technically be oriented to second use with similar physical properties (films, bags, pipes and fittings, profiles, and flexible sheets), provided an appropriate sorting and massification solution is identified.

This outcome is true only for SUA made of LDPE film as pure as possible. The result can be very different for multilayer films with various plastics.

Even if the technical feasibility is shown through this study, several challenges remain. The trade-off between the quality and quantity of the recycled plastic needs to be further evaluated. The cost and efficiency of sorting is another area that needs to be addressed.

The comparative LCA, based on a theoretical framework, shows that mechanical recycling for end of life presents the advantage of keeping material in the loop and does not show a significant statistical difference compared to incineration with regards to the climate change indicator.

The lack of accurate data (e.g., production volume, energy consumption, quantities of emissions), related to the emergence of this new circular scheme product, is one limit of this LCA. To assess the whole environmental impact, using different logistic scenarios to ensure the economic viability from a cost perspective, would require a more detailed cradle-to-cradle LCA, with more accurate data (raw materials, SUA production, location use, waste treatment streams), with relevant stakeholders, when the recycling scheme at scale will be defined.

**Supplementary Materials:** The following supporting information can be downloaded at: https://www.mdpi.com/article/10.3390/su142315557/s1. File S1: Tensile test results. File S2: Method of the LCA performed. File S3: Overall scope of the comparative LCA. File S4: Main assumptions & life cycle inventory. File S5: LCA results.

**Author Contributions:** Conceptualization, M.B. and G.D.; Methodology, G.D.; Validation, A.L.; Formal analysis, D.-N.L. and M.B.; Investigation, G.D.; Writing—original draft, D.-N.L., M.B. and G.D.; Writing—review & editing, M.B., G.D., K.M., E.D., A.L., J.-M.B. and C.-J.M.; Visualization, D.-N.L. and G.D.; Supervision, J.-M.B.; Project administration, M.B.; Funding acquisition, M.B. All authors have read and agreed to the published version of the manuscript.

**Funding:** This research received no external funding, and the article processing charge (APC) was funded by Sartorius.

**Institutional Review Board Statement:** Not applicable.

**Informed Consent Statement:** Not applicable.

**Data Availability Statement:** All the new data were presented in this article or in the Supplementary Materials.

**Conflicts of Interest:** The authors declare no conflict of interest.

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
