# Peer review of "Recycling of Post-Use Bioprocessing Plastic Containers—Mechanical Recycling Technical Feasibility"

_sustainability, doi:10.3390/su142315557_

Round 1

Reviewer 1 Report

The article is interesting and deals with an extremely topical issue. The text is recommended for publication with minor revisions, here are some points:

In the Methodology and testing section it is written that measurements with TGA and flexural strength were made, however, the curves or references to these tests do not appear in the text other than the result of the flexural tests, it would be useful to add them also in supporting information to get an idea of the behavior of the material.

A curiosity, avoiding sterilization and reprocessing, would it be possible, after pelletization, to be able to immobilize this plastic within concretes or mortars as a replacement of the natural aggregate?

In Figure 4 it would be better to put the y-axis in arbitrary units instead of percent transmittance. Also because some curves start at percentages higher than 100%.

Author Response

We would like to thank you for the comments. You will find enclosed the answers provided by the authors.

We remain available if required.

The authors.

Reviewer 2 Report

This paper presents a comprehensive study on the mechanical recycling of post-use bioprocessing plastic containers. The research topic well fits for the publishing scope of sustainability. However, I think the current version should be significantly improved to clarify the research highlights and avoid confusing for potential audience. Mainly, the main content of research results cannot be identified in abstract and introduction until Section 3 Materials and Methods. It seems that the LCA part should be the main concerns in this study. However, a lot of experimental results are presented to appeal the suitable properties of recycled plastics. Thus, I suggest major revision for this manuscript.

Major points:

1. The current abstract talks too much background. In consequence, the summary of research content and results are not so clear. For example, the authors only mentioned ‘characterize the materials generated by the recycling process, for suitability of further use’. I think the exact analysis should be explained, like the chemical structure and stability, mechanical properties were compared with virgin and common recycled plastic resins. Meanwhile, the quantitative results are favorable for illustrating research output. For example, the difference between recycled LDPE and the virgin one in Table 7. At last, a sentence to appeal the meaning or general applicability of research strategy can be added.

2. The introduction part gives a detailed overview about the environmental issues associated with the pharmaceutical industry. However, it is too far away from the main topic, plastic recycling. I suggest the authors to shorten the introduction of pharmaceutical industry and include more issues about plastic. For example, the environmental issues related the plastics like the microplastic in ecosystems, the problems of conventional treatments for plastic waste such as landfilling and incineration.

3. The Section 2 seems to be a supplementary introduction for the case study. Could this part move into supporting information with some critical information summarized in Section 1 or 3? Otherwise, the content is too much before reaching the research results.

4. The method part only presents the sample description, recycling processes and characterization methods. The LCA method is missing. Although the authors included the system boundary and data sources in the supporting information, they are lack of appropriate explanation. At least, a brief LCA interpretation should be presented in the main text.

5. The scope of LCA is clear. The authors want to compare the environmental impacts of mechanical recycling with incineration and chemical recycling. However, the life cycle inventory analysis included in the supporting information is confusing. Why the consumption of plastic resins differs for different treatments? How the recycled plastic reflects the life cycle model is uncertain. In addition, a company's environmental footprint was cited. However, how could this inventory data for a whole company can be used for a certain process or product?

6. The result and discussion part should be improved. Plenty of experimental data is presented to prove that the recycled LDPE is comparable with the virgin or commonly recycled ones. However, if the recycling is really feasible is more associated with the recycling system design and the demand of secondary plastic user, just as shown in Line 521-531. Thus, the strategy for recycling should be more strengthened and more frequently talked throughout the paper. In addition, only the overall value of life-cycle environmental impacts was presented in this study. The contribution of each process is uncertain. Combined with comment 5, the LCA part need significant revision.

7. The current conclusion is more like a future prospect part. The conclusion should be more concise and impactful. The detailed discussion of the shortcomings and future work should made in the previous section.

Minor suggestions:

8. Line 18: Did the bioprocess industry intend to use plastic for reducing environmental impacts?

9. Line 28-29: How ‘ downcycling” effect is limited ‘ is not well addressed. From the recycled material, the color indicates that it hard for using as the same packaging.

10. Line 30-31: Only said what was done. How about the quantitative results and key findings? And the value of research should be added after this sentence.

11. Line 40: The relationship between GHG and diseases is not clear with based on the current statement.

12. Line 81: ‘about 85% is general, non-hazardous waste’ If this 85% non-hazardous waste is mixed with 15% hazardous waste, it is also hard to recycle. Thus, a better waste management system is also important to be introduced here.

13. Line 88-91: The grammar of this sentence is strange.

14. Line 93: I think organic recycling is only applicable for bio-degradable plastics.

15. Line 131: Why LDPE? I think PVC is also commonly used.

16. Section 4: How to understand some analysis results for the suitability for recycling? For example, the FTIR just proved that it mainly included PE. Which one and how is calculated in Table 7 is unclear.

17. LCA results: Climate change short term, Do the short term mean IPCC GWP20a method or GWP100a?

18. Line 576: ‘similar physical properties to virgin material’ The results not support this conclusion. Close to common secondary plastic resin is more correct.

Author Response

THank you for the comments, we considered them all and tried to improve the papers by including modifications as much as possible. You will find enclosed the list of answers and modifications related to each comments.

We remain available if needed.

The authors.

Round 2

Reviewer 2 Report

The manuscript was well revised for publication.